# N-Acetylchitooligosaccharides Alleviate Pulmonary Inflammation and Modulate Glycerophospholipid Metabolism in Murine Acute Lung Injury

**DOI:** 10.3390/ijms26189128

**Published:** 2025-09-18

**Authors:** Xiongjie Sun, Fengnan Liu, Baifei Hu, Lin Zhu, Huabing Yang, Junping Zheng, Haiming Hu, Hongtao Liu

**Affiliations:** 1College of Basic Medical Sciences, Hubei University of Chinese Medicine, Wuhan 430065, China; sxj01072020@126.com (X.S.); 13756087984@163.com (F.L.); baifeihu3452@hbucm.edu.cn (B.H.); zhulin0107@163.com (L.Z.); huabingyang@hbucm.edu.cn (H.Y.); junpingzheng13@hbucm.edu.cn (J.Z.); 2School of Pharmacy, Hubei University of Chinese Medicine, Wuhan 430065, China; 3Hubei Shizhen Laboratory, Wuhan 430061, China; 4Key Laboratory of Traditional Chinese Medicine Resource and Compound Prescription, Ministry of Education, Hubei University of Chinese Medicine, Wuhan 430065, China; 5School of Laboratory Medicine, Hubei University of Chinese Medicine, Wuhan 430065, China

**Keywords:** pulmonary inflammation, untargeted metabolomics, metabolic reprogramming, glycerophospholipid metabolism, inflammatory mediators

## Abstract

N-acetylchitooligosaccharides (NACOS) are functional oligosaccharides derived from shrimp and crab shells that exhibit a variety of biological activities. This study investigates the protective effects of NACOS against acute lung injury (ALI) induced by lipopolysaccharides (LPS) in mice and explores its underlying metabolic regulatory mechanisms. Histopathological analysis showed that NACOS reduced pulmonary inflammation, edema, and disruption of tight junctions in ALI mice. Molecular analysis indicated that NACOS downregulated key inflammatory mediators, including *NLRP3*, *IL-1β*, *TNF-α*, *MPO*, and *GCSF*. Using untargeted metabolomics, glycerophospholipid metabolism was identified as the most significantly altered pathway following NACOS pre-treatment. Key regulated metabolites included triacylglycerols, phosphatidylethanolamines, lysophosphatidylcholines, and other glycerophospholipid derivatives. These findings suggest that NACOS exerts preventive effects through two primary mechanisms: the suppression of pro-inflammatory mediators and the modulation of glycerophospholipid metabolism. The identified metabolic alterations may serve as potential biomarkers for the progression of ALI and for monitoring prophylactic interventions.

## 1. Introduction

Acute lung injury (ALI) is a serious respiratory condition that can be triggered by a range of factors, both direct and indirect. It is characterized by increased permeability of endothelial cells, damage to alveolar epithelial tissue, infiltration of inflammatory cells, and reduced ventilatory function. In severe cases, ALI can progress to acute respiratory distress syndrome (ARDS), which has a mortality rate of up to 50% [1,2]. Currently, the management of ALI mainly involves respiratory support therapy, along with medications such as glucocorticoids and non-steroidal anti-inflammatory drugs [3]. However, these treatment options often show limited effectiveness in addressing the pathological and physiological changes associated with ARDS, especially in high-risk patients.

In recent years, metabolomics has become increasingly important for understanding the pharmacological mechanisms of various drugs. For instance, studies utilizing plasma metabolomics have shown that Poria cocos oligosaccharides can improve acute lung injury, potentially due to their impact on linoleic acid, linolenic acid, and arachidonic acid [4]. Additionally, lung tissue metabolomics has revealed that Huanglian Jiedu Tang inhibits inflammasome activation in ALI through the sphingolipid pathway [5]. Research indicates that ALI is frequently associated with significant metabolic disturbances, including abnormalities in amino acid [6], energy [7,8], and lipid metabolism [9,10]. Metabolomics offers a high-throughput, comprehensive, and intuitive approach for analyzing the dynamic metabolic changes that organisms experience in response to physiological, pathological, and pharmacological factors. This method effectively evaluates how drugs act on diseases, characterizes physiological or pathological states in vivo, assesses alterations in metabolic pathways, and identifies potential biomarkers. Consequently, it is a valuable tool for investigating the mechanisms underlying drug treatment for ALI.

Lipid metabolism is an essential biochemical process in the human body that involves the digestion, absorption, synthesis, and breakdown of fats. Lipids are crucial not only for energy storage and supply but also as fundamental structural components of various cellular membranes. Additionally, they act as signaling molecules that facilitate both intracellular and intercellular communication [11]. In the lungs, lipid metabolism is particularly active, and maintaining lipid homeostasis is vital for normal lung function. When lipid metabolism becomes dysregulated, it can lead to inflammation and significantly contribute to the development of ALI [12]. Recent advancements in metabolomics have identified notable changes in lipid metabolism associated with ALI, linking its metabolites to processes of inflammation and oxidative stress [5,13].

N-acetylchitooligosaccharides (NACOS), which are extracted from shrimp and crab shells, have been shown to possess a variety of biological activities [14]. For decades, NACOS have exhibited significant biological effects, including antitumor, immunostimulatory, and anti-inflammatory properties, as well as anti-β-amyloid activity in Alzheimer’s disease. Research suggests that using NACOS to intervene in the inflammatory response may be a promising strategy for preventing and treating clinical complications related to inflammation [15]. However, there is a notable gap in studies examining the protective mechanisms of NACOS against pulmonary inflammatory injury from a metabolomics perspective.

This study aims to use the LPS aerosol inhalation method to establish a mouse model of ALI. We will assess the pathological and physiological changes in lung tissues affected by injury and evaluate alterations in inflammatory mediators, pulmonary edema, and tight junctions. Furthermore, we will conduct a detailed analysis of plasma and lung tissue metabolomics to investigate the effects of NACOS on metabolic pathways during ALI and to identify potential metabolic markers. This research seeks to elucidate the protective mechanisms of NACOS in mice experiencing ALI.

## 2. Results

### 2.1. Effect of NACOS on LPS-Induce ALI in Mice

We examined the effects of NACOS on inflammation, edema, and tight junctions in the lung tissues of mice with ALI. H&E staining revealed that LPS stimulation led to neutrophil infiltration and widening of the alveolar septa. Pre-treatment with NACOS administration significantly reduced inflammation and alveolar wall thickening (Figure 1A). Meanwhile, quantitative assessment of lung injury based on established references demonstrated the histopathological impact of LPS or NACOS on lung tissues in ALI mice (Figure 1D). Widespread inflammatory damage induced by LPS was observed, accompanied by a significantly reduced injury score (*p* < 0.01 vs. Ctrl group). In contrast, prophylactic administration of NACOS resulted in a markedly higher score, indicating attenuated pathological changes (*p* < 0.01 vs. LPS group). Immunohistochemical analysis showed that LPS decreased the expression of γ-EnaC and Claudin-1 in the lungs. However, NACOS pre-treatment significantly restored the levels of these proteins (Figure 1B,C). The average optical density analysis of γ-EnaC and Claudin-1 using ImageJ 1.52a software also confirmed the effects of NACOS (Figure 1E,F). Alveolar lavage cell counting demonstrated a substantial increase in cell numbers in the LPS group, which was markedly suppressed by pre-treatment with NACOS (*p* < 0.01 vs. LPS group) (Figure 1D).

### 2.2. Effect of NACOS Treatment on Inflammatory Responses in Lung Tissues of ALI Mice

To thoroughly evaluate the inflammatory status in pulmonary tissues, we conducted comprehensive assessments of inflammatory mediator levels (Figure 2). The RT-qPCR analysis revealed that the LPS challenge significantly increased the expression of *NLRP3*, *IL-1β*, *TNF-α*, *IL-6*, *MCP-1*, *MPO*, *iNOS*, and *GCSF* in the lungs (*p* < 0.05 vs. Ctrl group). Importantly, NACOS administration prior to LPS substantially reversed these effects, resulting in significant downregulation of all measured inflammatory markers (*p* < 0.05 vs. LPS group).

### 2.3. Multivariate Statistical Analysis of Plasma Metabolomic Profiles

We conducted a comprehensive non-targeted metabolomic analysis of mouse plasma samples using liquid chromatography–mass spectrometry (LC-MS) in conjunction with the MetaboAnalyst 6.0 platform. This approach aimed to elucidate the metabolic basis underlying the anti-ALI effects of NACOS. To assess the intrinsic variation and quality control of our metabolomic data, we performed an unsupervised Principal Component Analysis (PCA) on the metabolic profiles prior to group classification. This initial analysis confirmed the technical reproducibility and data integrity, establishing a robust foundation for subsequent supervised multivariate analyses. The PCA results showed no significant clustering patterns in the endogenous metabolite profiles across the four experimental groups, including the LPS-induced ALI mice (Figure 3A).

To improve group discrimination, we subsequently performed supervised Partial Least Squares-Discriminant Analysis (PLS-DA). The PLS-DA model revealed clear metabolic separation between the Ctrl and LPS groups, confirming distinct metabolic disturbances induced by the LPS challenge. Notably, the NACOS + LPS group displayed a significant metabolic shift relative to the LPS group, with its profile closely resembling that of the Ctrl group (Figure 3B). This metabolic reversal suggests that pre-treatment with NACOS may restore LPS-induced metabolic dysregulation toward a more physiological state.

To further explore the metabolic perturbations, we constructed Orthogonal Projections to Latent Structures-Discriminant Analysis (OPLS-DA) models. Significant metabolic discrimination was observed between: Ctrl vs. LPS groups and LPS vs. NACOS + LPS groups, as indicated by distinct separation boundaries in the score plots (Figure 3C,D). These findings validate both the successful induction of ALI through LPS challenge and the metabolic restorative capacity of NACOS intervention.

A comprehensive analysis of plasma metabolomic profiles (positive/negative ion modes) revealed distinct alteration patterns. Volcano plot analysis demonstrated that the LPS challenge induced significant metabolic dysregulation, with 47 metabolites significantly upregulated and 66 downregulated compared to the Ctrl group (Figure 3E). However, pre-treatment with NACOS reversed this metabolic imbalance, resulting in 143 metabolites upregulated and 91 downregulated compared to the LPS group (Figure 3F).

### 2.4. Heatmap Analysis of Plasma Metabolic Profiles Across Experimental Groups

To systematically identify differential metabolites among the four experimental groups, we conducted comprehensive metabolite annotation and validation using multiple databases, including HMDB, KEGG, and MetaboAnalyst 6.0. We selected differential metabolites based on stringent criteria: a variable importance in projection (VIP) score greater than 1.0, a *p*-value less than 0.05, and fold change (FC) thresholds greater than 1.2 or less than 0.8. This analysis identified 239 significantly altered metabolites (Appendix A), which were subsequently subjected to pathway enrichment analysis using MetaboAnalyst 6.0.

The pathway analysis revealed 17 significantly perturbed metabolic pathways (Figure 4A), like glycerophospholipid metabolism, thiamine metabolism, D-amino acid metabolism, and glycosylphosphatidylinositol (GPI)-anchor biosynthesis. Notably, glycerophospholipid metabolism emerged as the most significantly altered pathway, suggesting its potential role as a central regulatory mechanism in the observed metabolic changes (Figure 4A).

To visualize metabolic alterations across experimental groups, we generated hierarchical clustering heatmaps focusing on the most significantly differential metabolites (Figure 4B). The heatmap analysis revealed distinct patterns of metabolic regulation following NACOS pre-treatment. NACOS upregulated metabolites such as γ-Glu-Leu, agomelatine, p-acetamidophenyl-β-D-glucuronide, and dehydrofelodipine. At the same time, it downregulated the levels of metabolites such as PE (22:6/24:1), TG (15:0/16:0/20:3), TG (14:0/18:3/18:1), 5,8,11-eicosatrienoic acid, decanoylcholine, and 6-O-acetyllaustroinulin.

### 2.5. NACOS Reversed Glycerophospholipid Metabolism in ALI Mice

As shown in Figure 4A, glycerophospholipid metabolism was identified as the most significantly enriched pathway in the plasma metabolome, both before and after the administration of NACOS. Consequently, we have illustrated the partial glycerophospholipid pathway diagram in Figure 5A, which includes pathways for TG, PE, LysoPE, PE-NMe2, LysoPC, MG, PA, and PG. In Figure 5B,I, the relative expression levels of individual glycerophospholipid metabolites among the four experimental groups are depicted.

The findings indicate that pre-treatment with NACOS administration significantly upregulated the expression of several metabolites, including MG (18:1/0:0/0:0), PG (18:2/18:1), PG (18:0/18:2), PE (P-16-0/22:4), LysoPE (0:0/14:0), and LysoPE (14:1/0:0). In contrast, NACOS pre-treatment led to a notable downregulation of TG (15:0/16:0/20:3), TG (14:0/18:3/18:1), MG (0:0/14:0/0:0), PE (22:6/24:1), PE (20:5/20:5), LysoPC (22:2), and PE-NMe2 (18:0/18:1).

### 2.6. Multivariate Statistical Analysis of Metabolites in Lung Tissue

Next, we utilized LC-MS to analyze non-targeted metabolites in the lung tissues of each experimental group. The results in Figure 6A indicate that the LPS group was significantly different from the other three groups, demonstrating a marked variation in endogenous metabolite profiles among the four groups, especially in the context of LPS-induced ALI. Using PLS-DA, we further distinguished the metabolite profiles, revealing clear differences between the Ctrl and LPS groups. Moreover, the NACOS + LPS group showed significant differences compared to the LPS group and aligned more closely with the Ctrl group’s profile, as shown in Figure 6B.

Further, we established the OPLS-DA model to study the lung metabolome profile. This model revealed distinct separation boundaries between the Ctrl and LPS groups, as well as between the LPS and LPS + NACOS groups (Figure 6C,D). These findings confirm the successful establishment of the LPS-induced ALI model and highlight the significant protective effect of NACOS.

A comprehensive analysis of positive and negative ion metabolites in the plasma of the four groups is presented. As illustrated in the volcano plot in Figure 6E, a total of 20 metabolites were upregulated and 19 metabolites downregulated in the LPS group compared to the Ctrl group. In comparison, the NACOS + LPS group had 94 metabolites upregulated and 61 downregulated relative to the LPS group (Figure 6F).

### 2.7. Heatmap Analysis of Metabolic Profiles in Lung Tissues of Experimental Groups

To identify the differential metabolites among the four groups in the lung metabolome, we conducted a robust cross-validation using various databases, including HMDB, KEGG, and MetaboAnalyst 6.0. Our selection criteria were VIP > 1, *p* < 0.05, FC > 1.2, or FC < 0.8. Ultimately, we identified 239 differential metabolites, as detailed in Appendix A. We plan to utilize MetaboAnalyst 6.0 to enrich and analyze the metabolic pathways associated with these metabolites. This analysis revealed 10 metabolic pathways in plasma, like Glycerophospholipid metabolism, Arachidonic acid metabolism, and Linoleic acid metabolism (Figure 7A). Notably, glycerophospholipid metabolism emerged as the most significantly affected pathway in lung tissues.

To visualize metabolic alterations across experimental groups, we generated hierarchical clustering heatmaps highlighting the most significant differential metabolites (Figure 7B). The heatmap analysis revealed distinct metabolic regulation patterns following NACOS pre-treatment. NACOS upregulated metabolites, such as Lansiumamide C, Leukotriene C4, and Pentacosanoylglycine. Meanwhile, NACOS downregulated the levels of metabolites, like Stearamide, PE-NMe2 (22:5/14:1), and PE-NMe2 (18:3/22:6).

### 2.8. NACOS Reversed Glycerophospholipid Metabolism in Lung Tissues of ALI Mice

As shown in Figure 8A, glycerophospholipid metabolism is the most significantly enriched pathway in the lung metabolome, both before and after the administration of NACOS. We illustrated a portion of the glycerophospholipid pathway in Figure 8A, which includes the following components: DG, PE, LysoPE, PE-NMe, PE-NMe2, PC, LysoPC, MG, and PGP.

The relative expression levels of individual glycerophospholipid metabolites among the four experimental groups are shown in Figure 8B–J. The results indicate that pre-treatment with NACOS significantly upregulated the expression of several metabolites, including DG (18:3/16:0/0:0), DG (13D5/13M5/0:0), DG (22:2n6/0:0/22:5n3), LysoPE (14:0/0:0), PE-NMe (22:6/20:0), PC (24:1/14:1), LysoPC (20:5), PE-NMe2 (18:3/22:4), and PGP (20:4/18:0). Conversely, NACOS pre-treatment resulted in a significant downregulation of PE (20:4/20:4), PE-NMe (20:3/22:5), PE-NMe2 (22:5/14:1), and PE-NMe2 (18:3/22:6).

### 2.9. Correlation Between Plasma and Lung Tissue Metabolites and Inflammatory Markers

To investigate the role of NACOS in regulating disorders of glycerophospholipid metabolism and its potential application in treating ALI, we employed Spearman correlation analysis to explore the relationship between changes in glycerophospholipid metabolites and inflammation-related indicators after NACOS administration.

As depicted in Figure 9A, mice pre-treated with NACOS displayed both positive and negative correlations between plasma glycerophospholipid metabolites and the expression of genes associated with inflammatory markers. Specifically, metabolites, such as TG (15:0/16:0/20:3), TG (14:0/18:3/18:1), and MG (0:0/14:0), showed significant positive correlations with inflammatory mediators like IL-1β, TNF-α, and IL-6. In contrast, several metabolites, like MG (18:1/0:0/0:0), PG (18:2/18:1), and PG (18:0/18:2), exhibited significant negative correlations only with IL-1β, without strong associations with other inflammatory mediators, including NLRP3, TNF-α, IL-6, MCP-1, MPO, iNOS, and GCSF.

In Figure 9B, glycerophospholipid metabolites, such as PE (20:4/20:4), PE-NMe (20:3/22:5), and PE-NMe2 (22:5/14:1), showed positive correlations with all eight inflammatory mediators, with most correlations being statistically significant. Additionally, diacylglycerols (DG (18:3/16:0/0:0), DG (13D5/13M5/0:0)), lysophospholipids (LysoPE (14:0/0:0), LysoPC (20:5)), and other glycerophospholipid species (PE-NMe (22:6/20:0), PC (24:1/14:1), PE-NMe2 (18:3/22:4), and PGP (20:4/18:0)) were found to have significant negative correlations with all eight inflammatory mediators mentioned above.

## 3. Discussion

This study utilized LPS to create an acute lung injury (ALI) mouse model. Our current study was primarily designed to investigate the early protective mechanisms of NACOS, focusing on the initial inflammatory burst that occur within the first hours after LPS challenge. The 8-hour time point was selected to optimally capture these key early events, as suggested by previous literature [16,17]. Histological analysis with H&E staining showed that NACOS significantly reduced LPS-induced lung injury in mice by decreasing inflammatory cell infiltration and preventing the thickening of alveolar walls. Pulmonary edema is a crucial indicator of ALI [18,19], and the disruption of tight junctions is a commonly observed phenomenon during such injuries [20,21]. Immunoblotting analysis of γ-EnaC and Claudin-1 confirmed that administering NACOS considerably improved the symptoms of lung injury in the ALI mice. Additionally, we performed an alveolar lavage and found that the total cell counting in the lavage fluid from the LPS group were significantly elevated, indicating severe lung injury. Pre-treatment with NACOS effectively reversed this trend and contributed to the alleviation of lung injury.

Neutrophils, macrophages, endothelial cells, and alveolar epithelial cells are recognized as key contributors to the pathogenesis of ALI. These cells release a variety of pro-inflammatory factors, including *NLRP3*, *IL-1β*, *TNF-α*, and *IL-6* [22,23]. Additionally, factors such as *MCP-1*, *MPO*, *iNOS*, and *GCSF* promote the migration and infiltration of inflammatory cells to the site of inflammation, which can lead to the development of various diseases [24,25,26,27,28]. In this study, the over-expression of these pro-inflammatory factors in lung tissues of ALI mice was remarkably suppressed by NACOS administration, indicating that NACOS has the potential to alleviate pulmonary injury by inhibiting inflammatory responses in ALI mice.

In metabolomics studies of ALI, blood and lung tissue are recognized as the most suitable specimens for analysis. The lungs are the primary organs affected in ALI, while blood serves as an optimal medium for housing a diverse range of small molecule metabolites [29]. Changes in blood composition can indicate both normal and abnormal variations in the body’s organs. In this experiment, we identified and analyzed endogenous metabolites in the plasma and lung tissue of LPS-induced ALI mice pre-treated with NACOS. A total of 239 and 56 differentially abundant metabolites were identified in plasma and lung tissue, respectively. These metabolites were significantly enriched in a diverse array of metabolic pathways, including glycerophospholipid metabolism, thiamine metabolism, D-amino acid metabolism, glycosylphosphatidylinositol (GPI)-anchor biosynthesis, purine metabolism, terpenoid backbone biosynthesis, ether lipid metabolism, steroid hormone biosynthesis, sphingolipid metabolism, glyoxylate and dicarboxylate metabolism, cysteine and methionine metabolism, glycine, serine and threonine metabolism, drug metabolism—other enzymes, tryptophan metabolism, amino sugar and nucleotide sugar metabolism, arachidonic acid metabolism, and drug metabolism—cytochrome P450. Notably, glycerophospholipid metabolism emerged as the most significantly altered pathway in both plasma and lung tissue metabolomes.

Previous studies have demonstrated that phospholipid metabolism is significantly affected during ALI [30]. Phospholipids, the primary constituents of cell membranes, can undergo oxidation and modification through enzymatic and non-enzymatic pathways, leading to the formation of oxidized phospholipid products such as sphingomyelins and glycerophospholipids. These oxidized products have markedly different polarity and structure compared to their parent forms, contributing to their functional diversity and allowing them to play dual roles in promoting or inhibiting inflammation in lung injury conditions [31,32,33,34]. Among these metabolites, glycerophospholipids can be hydrolyzed by phospholipase A2, resulting in a series of biochemical transformations that produce various biologically relevant phospholipids [35]. In our study, several metabolites like TG (15:0/16:0/20:3) and TG (14:0/18:3/18:1) were found to be significantly upregulated in the ALI model group of mice, with these changes being reversed by NACOS pre-treatment. Additionally, other phospholipids such as MG (18:1/0:0/0:0) and PG (18:2/18:1) were significantly downregulated in ALI mice following pre-treatment with NACOS. These findings suggest that NACOS can modulate the disruption of the glycerophospholipid metabolism pathway in ALI mice in both directions.

We propose that there is a reciprocal relationship between NACOS, inflammatory mediators, and glycerophospholipid metabolism, considering the regulatory effects of NACOS on these factors. To investigate this connection, we conducted a Spearman correlation analysis. Our results showed that several glycerophospholipid metabolites, which were significantly altered, exhibited both positive and negative correlations with eight key inflammatory mediators across the four experimental groups. In summary, we propose that NACOS may help prevent ALI by modulating the glycerophospholipid metabolism pathway.

As we reported, the profound suppression of pro-inflammatory factors in lung tissue homogenates strongly suggests that NACOS modulates the functional state of macrophages residing in the lungs (Figure 2). Meanwhile, our untargeted metabolomics data provide compelling circumstantial evidence that alveolar macrophages (AMs) may be a key cellular target of NACOS. This interpretation is supported by the finding that glycerophospholipid metabolism was the most significantly altered pathway [36,37] (Fig. 4A and 7A). Furthermore, perturbations in other immunometabolic pathways, including arachidonic acid metabolism [38,39] and key amino acid metabolism pathways [40,41,42] (such as glycine, serine, threonine, cysteine, and methionine), also suggest that NACOS may promote a shift in macrophage polarization toward a pro-resolving or homeostatic phenotype by altering lipid mediator production and enhancing antioxidant capacity. Changes in specific metabolites—such as LysoPC(22:2), LysoPC(20:5), leukotriene C4, MG(18:1/0:0/0:0), PG(18:2/18:1), PG(18:0/18:2), PE(P-16:0/22:4), LysoPE(0:0/14:0), and LysoPE(14:1/0:0)—further support this explanation. Therefore, although direct validation is currently lacking, we propose that NACOS alleviates ALI, at least in part, by reprogramming AM lipid metabolism and function. Future studies employing single-cell transcriptomics or AM-specific metabolomics will be conducted to validate this mechanism.

It is important to note that our study primarily focused on the early acute phase of ALI. While this allowed us to delineate the initial protective mechanisms of NACOS against inflammatory, it does not address its potential effects on the later phases of the disease, including resolution of inflammation and tissue repair. Future studies employing longer time courses, extending up to 14 days, will be essential to determine whether NACOS merely delays or truly prevents lung injury, and to assess its impact on critical long-term outcomes such as pulmonary fibrosis and survival. Another limitation of the present study is the use of male mice, a decision made to first establish a clear and stable model of ALI while controlling for potential confounding variables, including the hormonal fluctuations associated with the female estrous cycle [43]. We acknowledge that this choice limits the generalizability of our findings. In addition, we acknowledge that the current study employed a prophylactic administration regimen to evaluate the protective potential of NACOS against ALI. While this approach is valuable for establishing proof-of-concept and elucidating underlying mechanisms, it does not fully recapitulate the clinical scenario in which treatment is initiated after the manifestation of symptoms. Nevertheless, the potent anti-inflammatory and tissue-reparative effects observed—such as reduction in inflammatory cell infiltration, suppression of pro-inflammatory cytokines, restoration of alveolar architecture, and upregulation of key proteins involved in epithelial barrier function (Claudin-1) and alveolar fluid clearance (γ-EnaC)—strongly suggest that NACOS may also possess therapeutic efficacy in established ALI. These properties are essential for mitigating ongoing injury and facilitating recovery in clinical ALI/ARDS. Furthermore, the broad pharmacological profile of NACOS, including its documented immunomodulatory, anti-inflammatory, and barrier-protective activities, supports its potential applicability as a therapeutic agent beyond prophylaxis. Future studies utilizing post-exposure interventional designs will be critical to definitively establish its efficacy in reversing fully manifested lung injury.

## 4. Materials and Methods

### 4.1. Chemicals and Reagents

NACOS with a polymerization degree of 2–6 was prepared as previously reported [44]. Lipopolysaccharides (LPS) from *Escherichia coli* 055: B5 were obtained from Amresco (Solon, OH, USA). TRIzol Reagent, RNA Reverse Transcription Kit, and SYBR QPCR mixture were purchased from Summer Biotechnology Co., Ltd. (Beijing, China). Antibodies against γ-EnaC and Claudin-1 were obtained from Cell Signaling Technology (Danvers, MA, USA). All other chemicals were sourced from Sinopharm Chemical Reagent Co., Ltd. (Shanghai, China).

### 4.2. Animal Experiment

Male BALB/c mice weighing 20–22 g were purchased from the Hubei Provincial Center for Disease Control and Prevention. The mice were acclimatized for one week before the experiment. The housing conditions were maintained at appropriate temperature and humidity levels, with continuous access to food and water, and a 12-h light-dark cycle.

The mice were randomly divided into four groups, each containing 12 mice: a control group (Ctrl), an LPS group (LPS), an LPS + NACOS group (LPS + NACOS), and a NACOS group (NACOS). An ALI mouse model was induced using LPS (0.5 mg/mL) via inhalation at a flow rate of 0.8 mL/min for 20 min. The Ctrl and NACOS groups received saline at the same flow rate and dosage. For two weeks prior to modeling, the mice were allowed to freely drink a NACOS solution at a concentration of 1 mg/mL. Eight hours after the LPS challenge, the mice were anesthetized. The left lung was used to collect bronchoalveolar lavage fluid (BALF). The superior lobe of the right lung was fixed in paraformaldehyde, and the remaining lung tissue was frozen in liquid nitrogen and stored at −80 °C. Animal experiments were performed following the Ethical Experimentation Committee of the Hubei University of Chinese Medicine (permission number: SYXK-2017-0067) and the National Act on the Use of Experimental Animals.

### 4.3. Histopathology Analysis of Lung Tissues

The fixed lung tissue was embedded in paraffin and sectioned into 3 μm thick slices. The sections were stained with hematoxylin & eosin and subsequently evaluated using a light microscope (Leica; Wetzlar, Germany). Lung injury was scored based on the severity of four histological features: interstitial edema, hemorrhage, inflammatory cell infiltration, and alveolar septal widening. Each feature was assigned a score from 0 to 4, where 0 indicated no injury, 1 represented injury in 25% of the field, 2 for 50%, 3 for 75%, and 4 indicated injury throughout the entire field. The total score was obtained by summing the points from all four indicators.

### 4.4. Bronchoalveolar Lavage Fluid (BALF) Collection and Cell Counting

The mice were anesthetized with ether, and an incision was made in the neck. An intravenous indwelling needle was then inserted into the exposed trachea. The right lung was ligated, and 0.6 mL of normal saline was infused into the left lung. After repeated aspiration, the lavage fluid was collected. The BALF was centrifuged at 1500 rpm for 5 min, and the sediment was collected. This sediment was resuspended in PBS, and the cells in suspension were counted using a hemocytometer.

### 4.5. RNA Extraction and Real-Time Quantitative PCR (RT-qPCR)

Total RNA was extracted using TRIzol reagents according to the manufacturer’s protocol (Summerbio, Beijing, China). For RT-qPCR, cDNA was synthesized using a first-strand cDNA synthesis kit (Summerbio, Beijing, China). The relative mRNA levels of NLRP3, IL-1β, TNF-α, IL-6, MCP-1, MPO, iNOS, GCSF, and β-actin were measured using the CFX Connect Real-Time System (Bio-Rad, Hercules, CA, USA) with SYBR Green I Master PCR Mix (Summerbio, Beijing, China). The amplification protocol was as follows: 95 °C for 10 min; followed by 40 amplification cycles (95 °C for 10 s, 60 °C for 30 s). The primer sequences utilized in this study are summarized in Table 1.

### 4.6. Statistical Analysis

Data were presented as means ± standard deviation (SD). Statistical analyses were conducted using GraphPad Prism version 7.0 (GraphPad Software, San Diego, CA, USA). To compare the experimental groups, analysis of variance (ANOVA) was applied, followed by Tukey–Kramer post hoc analysis. A *p*-value of less than 0.05 was considered statistically significant.

## 5. Conclusions

In this study, we developed a mouse model of ALI. Our findings imply that NACOS significantly protects against pulmonary edema, lung barrier disruption, and inflammation associated with ALI. Metabolomic analyses revealed distinct differences between the Ctrl and LPS groups, particularly highlighting notable alterations in the glycerophospholipid metabolic pathway following NACOS pre-treatment. Key regulated metabolites included various triacylglycerols, phosphatidylethanolamines, lysophosphatidylcholines, and diacylglycerols. These results suggest that NACOS pre-treatment initiates its prophylactic effect through dual mechanisms: the suppression of pro-inflammatory mediators and the modulation of glycerophospholipid metabolism, which may serve as potential biomarkers for the progression of ALI and for protective monitoring.

## Figures and Tables

**Figure 1 ijms-26-09128-f001:**
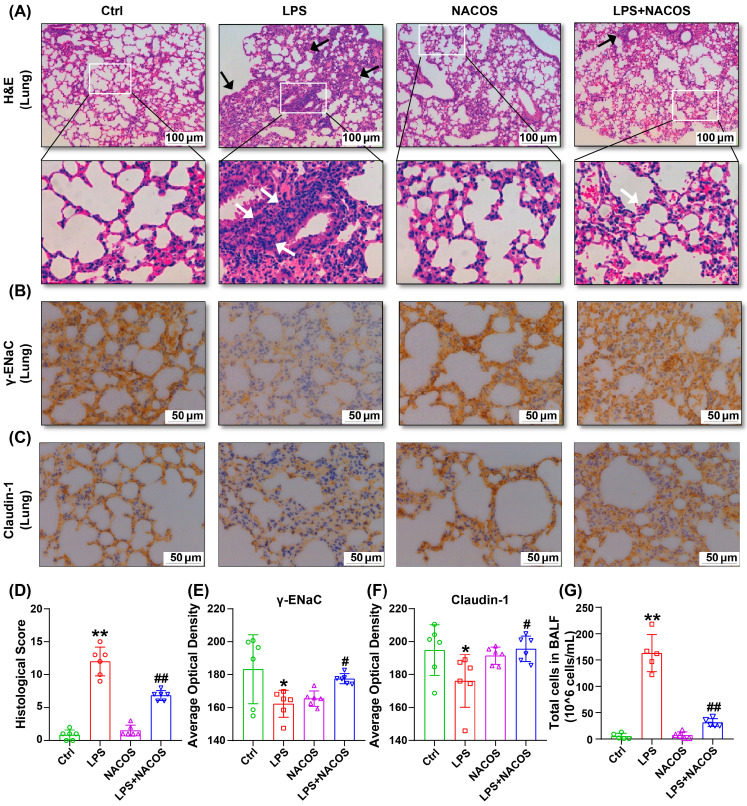
NACOS-reduced lung injury in ALI mice. The mice were randomly divided into four groups, each containing 12 mice: a control group (Ctrl), an LPS group (LPS), an LPS + NACOS group (LPS + NACOS), and a NACOS group (NACOS). Mice were pretreated with NACOS (1 mg/mL in drinking water) for two weeks prior to induction of ALI by LPS inhalation (0.5 mg/mL, 20 min). After the experiment, all mice were euthanized and lung tissue was collected for histological analysis. (**A**) Lung tissues from all groups were collected, and representative slides of H&E staining are shown. White arrows indicate inflammatory cells, while black arrows highlight thicker alveolar walls. (**B**) Expression of γ-ENaC in lung tissues assessed by immunohistochemistry. (**C**) Expression of Claudin-1 in lung tissues evaluated by immunohistochemistry. (**D**) The pathology injury score for each group was based on the assessment of interstitial edema, hemorrhage, inflammatory cell infiltration in the alveoli, and alveolar septal widening. The scores corresponded to the extent of injury: 0 (none), 1 (25% of the field), 2 (50%), 3 (75%), and 4 (diffuse injury). The total score was obtained by summing the points from all four indicators. (**E**) Average optical density analysis of γ-EnaC was performed using ImageJ software. (**F**) Average optical density analysis of Claudin-1 was performed using ImageJ software. (**G**) Bronchoalveolar lavage fluid (BALF) was centrifuged and resuspended for cell counting. All the results are presented as Mean ± SD. * *p* < 0.05, ** *p* < 0.01 vs. Ctrl group, # *p* < 0.05, ## *p* < 0.01 vs. LPS group.

**Figure 2 ijms-26-09128-f002:**
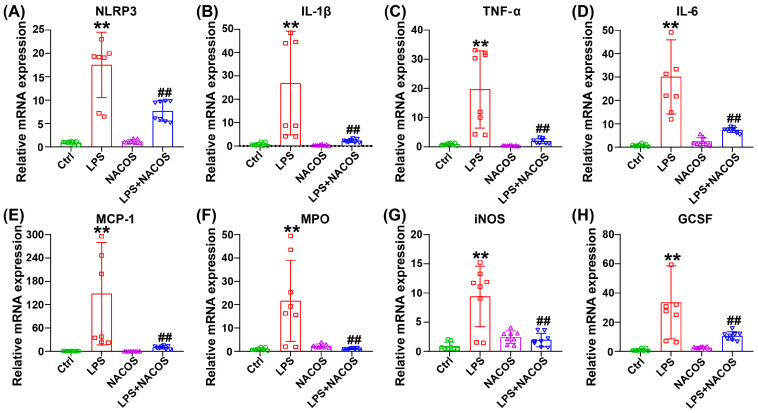
NACOS-reduced release of inflammatory mediators in lung tissue of mice with ALI. The mice were randomly divided into four groups, each containing 12 mice: a control group (Ctrl), an LPS group (LPS), an LPS + NACOS group (LPS + NACOS), and a NACOS group (NACOS). Mice were pretreated with NACOS (1 mg/mL in drinking water) for two weeks prior to induction of ALI by LPS inhalation (0.5 mg/mL, 20 min). After the experiment, all mice were euthanized and lung tissue was collected for further analysis. The mRNA expression levels of various pro-inflammatory markers, including *NLRP3* (**A**), *IL-1β* (**B**), *TNF-α* (**C**), *IL-6* (**D**), *MCP-1* (**E**), *MPO* (**F**), *iNOS* (**G**), and *GCSF* (**H**), which were quantified using RT-qPCR. All results are expressed as Mean ± SD. All the results are presented as Mean ± SD. ** *p* < 0.01 vs. Ctrl group, ## *p* < 0.01 vs. LPS group.

**Figure 3 ijms-26-09128-f003:**
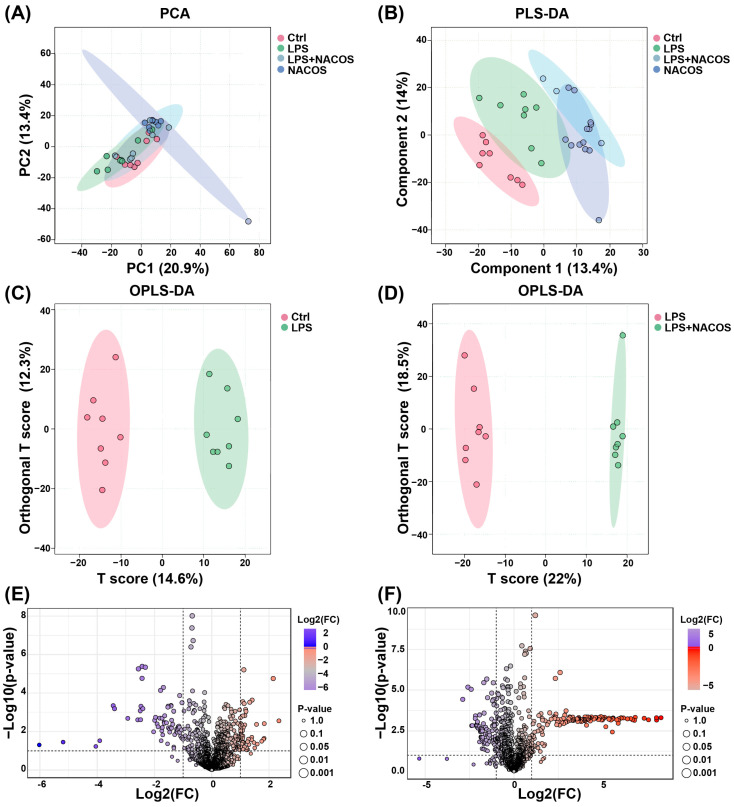
NACOS-regulated plasma metabolome in ALI mice. The mice were randomly divided into four groups, each containing 12 mice: a control group (Ctrl), an LPS group (LPS), an LPS + NACOS group (LPS + NACOS), and a NACOS group (NACOS). Mice were pretreated with NACOS (1 mg/mL in drinking water) for two weeks prior to induction of ALI by LPS inhalation (0.5 mg/mL, 20 min). After the experiment, all mice were euthanized and blood samples were collected for metabolomics analysis. (**A**,**B**) Metabolite analyses of Principal Component Analysis (PCA) (**A**) and Partial Least Squares-Discriminant Analysis (PLS-DA) (**B**) scores of plasma samples from each group. (**C**,**D**) Orthogonal Projections to Latent Structures-Discriminant Analysis (OPLS-DA) score plots (**C**) comparing the Ctrl and LPS groups, and (**D**) comparing the LPS group with the LPS + NACOS group. (**E**,**F**) Volcano plots (**E**) showing metabolites between the Ctrl group and the LPS group, and (**F**) showing metabolites between the LPS group and the LPS + NACOS group.

**Figure 4 ijms-26-09128-f004:**
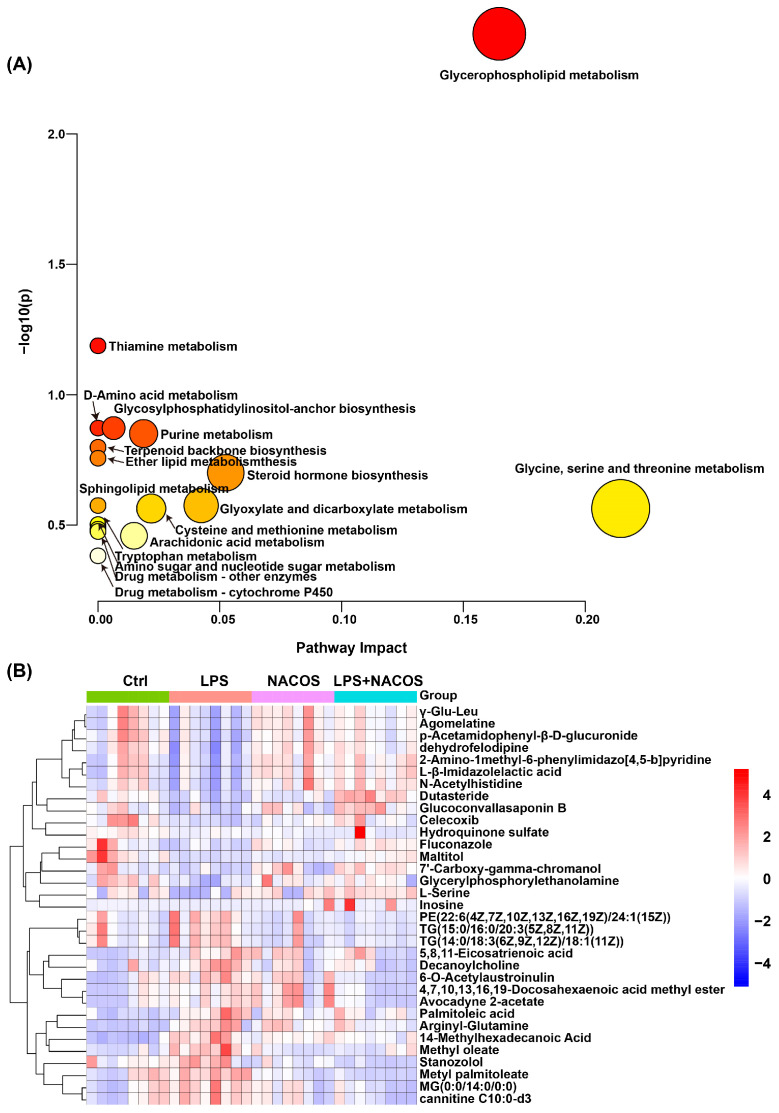
Differential metabolic pathways and enrichment analysis of differential metabolites in plasma metabolome of ALI mice. The mice were randomly divided into four groups, each containing 12 mice: a control group (Ctrl), an LPS group (LPS), an LPS + NACOS group (LPS + NACOS), and a NACOS group (NACOS). Mice were pretreated with NACOS (1 mg/mL in drinking water) for two weeks prior to induction of ALI by LPS inhalation (0.5 mg/mL, 20 min). After the experiment, all mice were euthanized and blood samples were collected for metabolomics analysis. (**A**) Differential metabolite enrichment pathway. The y-axis represents the statistical significance (−log10(*p*-value)) of the pathway enrichment. The x-axis represents the pathway impact value, which indicates the centrality of the matched metabolites within the pathway topology. The size and color gradient of each bubble are proportional to the pathway impact value. The arrows indicate the corresponding metabolic pathways. (**B**) Enrichment heatmap of 33 significantly different metabolites. The color scale is proportional to the relative abundance, ranging from blue (low) to red (high).

**Figure 5 ijms-26-09128-f005:**
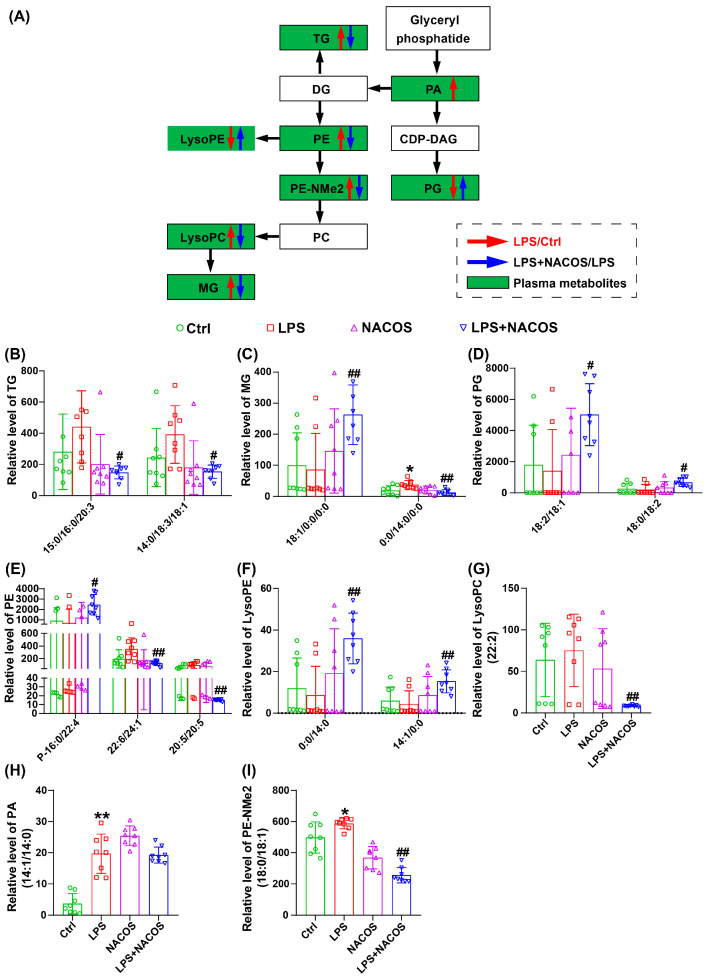
Alterations in glycerophospholipid metabolic profile in the plasma of ALI mice. The mice were randomly divided into four groups, each containing 12 mice: a control group (Ctrl), an LPS group (LPS), an LPS + NACOS group (LPS + NACOS), and a NACOS group (NACOS). Mice were pretreated with NACOS (1 mg/mL in drinking water) for two weeks prior to induction of ALI by LPS inhalation (0.5 mg/mL, 20 min). After the experiment, all mice were euthanized and blood samples were collected for metabolomics analysis. (**A**) Schematic representation of glycerophospholipid metabolic pathways affected by LPS and NACOS. Red arrows denote up-regulation in LPS group compared to Ctrl group, while blue arrows denote down-regulation in LPS + NACOS group compared to LPS group. (**B**) The relative level of TG among the four experimental groups. (**C**) The relative level of MG among the four experimental groups. (**D**) The relative level of PG among the four experimental groups. (**E**) The relative level of PE among the four experimental groups. (**F**) The relative level of LysoPE among the four experimental groups. (**G**) The relative level of LysoPC (22:2) among the four experimental groups. (**H**) The relative level of PA (14:1/14:0) among the four experimental groups. (**I**) The relative level of PE-NMe2 (18:0/18:1) among the four experimental groups. The bar graphs indicate significant changes in specific glycerophospholipid species. All the results are presented as Mean ± SD. * *p* < 0.05, ** *p* < 0.01 vs. Ctrl group, # *p* < 0.05, ## *p* < 0.01 vs. LPS group.

**Figure 6 ijms-26-09128-f006:**
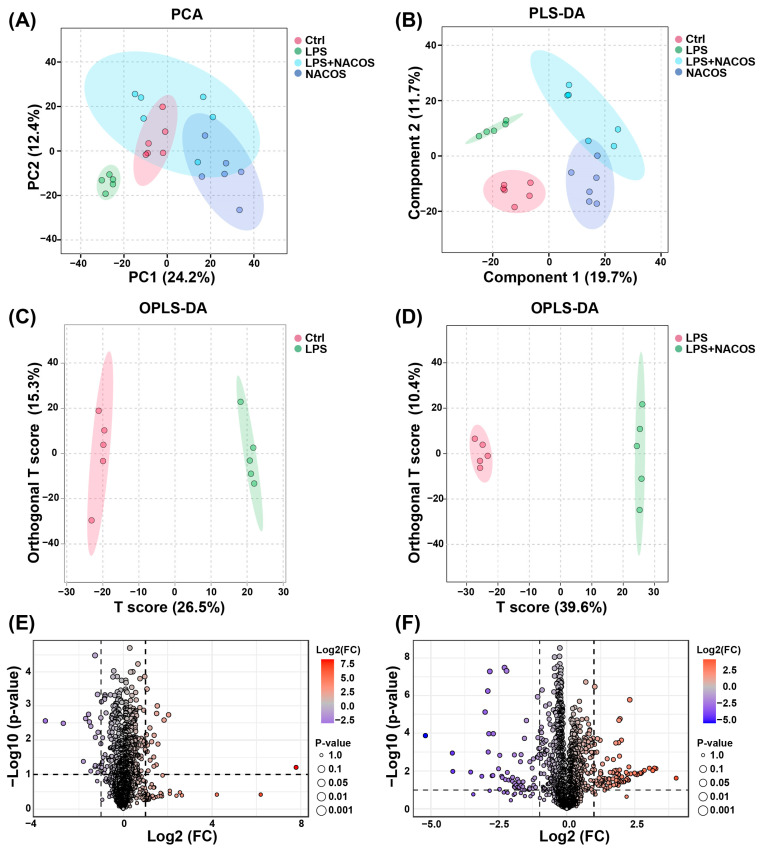
NACOS-regulated lung metabolome in ALI mice. The mice were randomly divided into four groups, each containing 12 mice: a control group (Ctrl), an LPS group (LPS), an LPS + NACOS group (LPS + NACOS), and a NACOS group (NACOS). Mice were pretreated with NACOS (1 mg/mL in drinking water) for two weeks prior to induction of ALI by LPS inhalation (0.5 mg/mL, 20 min). After the experiment, all mice were euthanized and lung tissue samples were collected for metabolomics analysis. (**A**,**B**) Metabolite analysis using Principal Component Analysis (PCA) (**A**) and Partial Least Squares Discriminant Analysis (PLS-DA) (**B**), comparing lung tissues from each experimental group. (**C**) Orthogonal PLS-DA (OPLS-DA) score plot highlights the differences between the Ctrl and LPS groups. (**D**) Orthogonal PLS-DA (OPLS-DA) score plot shows the comparison between the LPS and LPS + NACOS groups. (**E**) Volcano plot of the metabolite differences between the Ctrl and LPS groups. (**F**) Volcano plot of metabolite differences between the LPS and LPS + NACOS groups.

**Figure 7 ijms-26-09128-f007:**
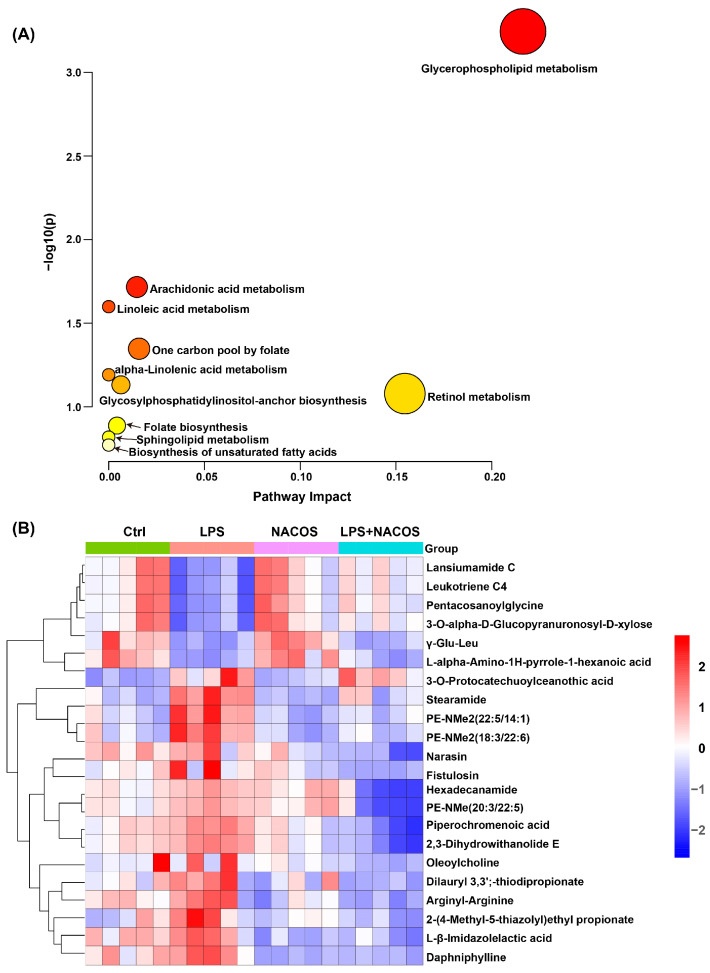
Differential metabolic pathways and enrichment analysis of metabolites in the lung metabolome of ALI mice. The mice were randomly divided into four groups, each containing 12 mice: a control group (Ctrl), an LPS group (LPS), an LPS + NACOS group (LPS + NACOS), and a NACOS group (NACOS). Mice were pretreated with NACOS (1 mg/mL in drinking water) for two weeks prior to induction of ALI by LPS inhalation (0.5 mg/mL, 20 min). After the experiment, all mice were euthanized and lung tissue samples were collected for metabolomics analysis. (**A**) Pathway enrichment analysis of differential metabolites in lung tissue. The y-axis represents the statistical significance (-log10(*p*-value)) of the pathway enrichment. The x-axis represents the pathway impact value, which indicates the centrality of the matched metabolites within the pathway topology. The size and color gradient of each bubble are proportional to the pathway impact value. The arrows indicate the corresponding metabolic pathways. (**B**) Heatmap displaying the enrichment of 22 significantly different metabolites in lung tissue. The color scale is proportional to the relative abundance, ranging from blue (low) to red (high).

**Figure 8 ijms-26-09128-f008:**
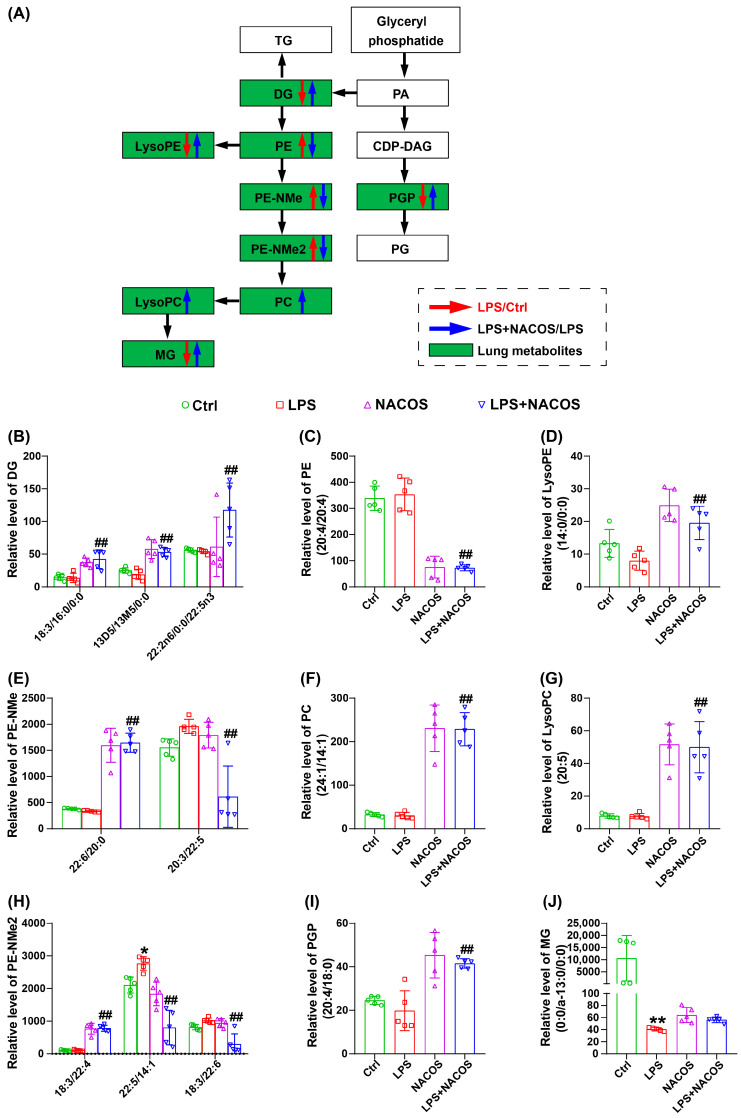
Glycerophospholipid metabolic profiles in the lungs of ALI mice. The mice were randomly divided into four groups, each containing 12 mice: a control group (Ctrl), an LPS group (LPS), an LPS + NACOS group (LPS + NACOS), and a NACOS group (NACOS). Mice were pretreated with NACOS (1 mg/mL in drinking water) for two weeks prior to induction of ALI by LPS inhalation (0.5 mg/mL, 20 min). After the experiment, all mice were euthanized and lung tissue samples were collected for metabolomics analysis. (**A**) Red arrows denote up-regulation in LPS group compared to Ctrl group, while blue arrows denote down-regulation in LPS + NACOS group compared to LPS group. (**B**) The relative level of DG among the four experimental groups. (**C**) The relative level of PE (20:4/20:4) among the four experimental groups. (**D**) The relative level of LysoPE (14:0/0:0) among the four experimental groups. (**E**) The relative level of PE-NMe among the four experimental groups. (**F**) The relative level of PC (24:1/14:1) among the four experimental groups. (**G**) The relative level of LysoPC (20:5) among the four experimental groups. (**H**) The relative level of PE-NMe2 among the four experimental groups. (**I**) The relative level of PGP (20:4/18:0) among the four experimental groups. (**J**) The relative level of MG (0:0/a–13:0/0:0) among the four experimental groups. All the results are presented as Mean ± SD. * *p* < 0.05, ** *p* < 0.01 vs. Ctrl group, ## *p* < 0.01 vs. LPS group.

**Figure 9 ijms-26-09128-f009:**
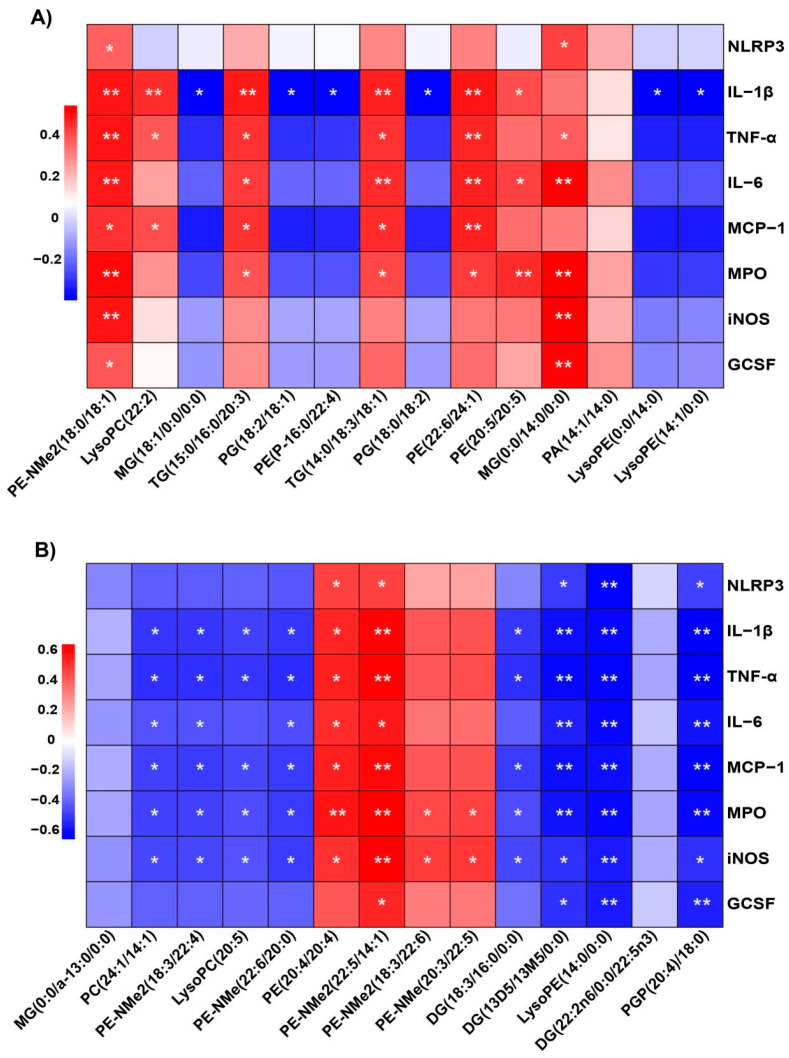
Correlations between alterations in metabolites observed in plasma and lung tissue, along with associated changes in inflammatory markers in ALI mice. The analysis utilized the Spearman correlation method. (**A**) Relationship between altered metabolites in plasma and changes in physiochemical indices. The color scale is proportional to the relative abundance, ranging from blue (low) to red (high). (**B**) Correlation between modified metabolites in lung tissues and changes in physiochemical indices. The color scale is proportional to the relative abundance, ranging from blue (low) to red (high). * *p* < 0.05 and ** *p* < 0.01.

**Table 1 ijms-26-09128-t001:** Primer sequences for RT-qPCR.

Gene	Primer Name	Sequence (5′→3′)
*NLRP3*	NLRP3-F	ATTACCCGCCCGAGAAAGG
	NLRP3-R	TCGCAGCAAAGATCCACACAG
*IL-1β*	IL-1β-F	GGCTGGACTGTTTCTAATGC
	IL-1β-R	ATGGTTTCTTGTGACCCTGA
*TNF-α*	TNF-α-F	GGGTGTTCATCCATTCTC
	TNF-α-R	GGAAAGCCCATTTGAGT
*IL-6*	IL-6-F	GAAACCGCTATGAAGTTCCTCTCTG
	IL-6-R	TGTTGGGAGTGGTATCCTCTGTGA
*MCP-1*	MCP-1-F	TGACCCCAAGAAGGAATGGG
	MCP-1-R	ACCTTAGGGCAGATGCAGTT
*MPO*	MPO-F	GCCCACCGAATGACAAG
	MPO-R	GAAGCCATTGCGATTGA
*iNOS*	iNOS-F	TTCAGTATCACAACCTCAGCAAG
	iNOS-R	TGGACCTGCAAGTTAAAATCCC
*GCSF*	GCSF-F	ATGGCTCAACTTTCTGCCCAG
	GCSF-R	CTGACAGTGACCAGGGGAAC
*β-actin*	β-actin-F	CGTGAAAAGATGACCCAGA
	β-actin-R	GTCCATCACAATGCCTGT

## Data Availability

Data will be made available on request.

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
