# Peer review of "N-Acetylchitooligosaccharides Alleviate Pulmonary Inflammation and Modulate Glycerophospholipid Metabolism in Murine Acute Lung Injury"

_ijms, 2025, doi:10.3390/ijms26189128_

Round 1
Reviewer 1 Report
Comments and Suggestions for Authors
The manuscript describes the use of a novel oligosaccharide derived from shell fish to reduce acute lung injury in mice. ALI has a number of causes. The risk of being untreated is progression to ARDS with a significant risk of mortality.
Line 74/75 – Given the high incidence of shell fish allergy globally is there a risk of anaphylaxis from NACOS? If not, what level of purification is required to eliminate the risk of potential allergen contamination during isolation and NACOS manufacture.
Line 95 and 96 – Looking at the methods this is an LPS challenge approach. The reduced thickening of the epithelium is consequently a protective mechanism. As a research project it is understood that a experimental design doesn’t have to follow a clinical application. However, in a clinical setting presumably patients present with ALI or ARDS. Do the authors believe this approach will reverse of stabilize disease, when treatment is delivered after the condition is shown to exist? Some discussion of the clinical use of the drug product should be included.
Lin 196 and197 - What are the upregulated metabolites indicative of mechanistically? Does their upregulation also correlate with recovery of the lungs?
Line 330 – Since alveolar macrophages play a significant role in phospholipid metabolism were they looked at as a unique cell type for the impact of NACOS? In all of the metabolomic studies were specific markers associated with macrophage activity?
Author Response
Dear Editor,
On behalf of all the contributing authors, I highly appreciate your letter and reviewers’ constructive comments concerning our article entitled “Glycerophospholipid metabolism modulation by NACOS alleviates pulmonary inflammation in murine acute lung injury” (ID: ijms-3808091). These comments are very pertinent and valuable for improving our paper. According to the reviewer’s suggestions, we have revised the manuscript. The revision details were marked in red. If you have other suggestions for mending the paper, we would greatly appreciate your comments. The responses to reviewers’ comments are listed below.
Comments from the editors and reviewers:
-Reviewer 1
Reviewer #1: The manuscript describes the use of a novel oligosaccharide derived from shell fish to reduce acute lung injury in mice. ALI has a number of causes. The risk of being untreated is progression to ARDS with a significant risk of mortality.
Comment 1. Line 74/75 – Given the high incidence of shell fish allergy globally is there a risk of anaphylaxis from NACOS? If not, what level of purification is required to eliminate the risk of potential allergen contamination during isolation and NACOS manufacture.
Response: Thank you for your suggestions. The reviewer raises a valid concern regarding potential allergenicity. We would like to clarify that the identified allergenic components in shellfish mainly include tropomyosin, arginine kinase, myosin light chain, and sarcoplasmic calcium binding protein (DOI: 10.1111/all. 13115). In contrast, NACOS is a low molecular weight oligosaccharide derived from chitin—a polysaccharide—through enzymatic hydrolysis by chitinase. It is crucial to note that no allergic reactions to NACOS itself have been reported in the literature. Given its distinct biochemical nature (a carbohydrate versus a protein) and its production process that eliminates proteinaceous material, NACOS is not considered a shellfish allergen and falls outside the scope of known allergenic compounds from these sources (DOI: 10.1111/j.1574-6968.2002.tb11108.x).
Comment 2. Line 95 and 96 – Looking at the methods this is an LPS challenge approach. The reduced thickening of the epithelium is consequently a protective mechanism. As a research project it is understood that an experimental design doesn’t have to follow a clinical application. However, in a clinical setting presumably patients present with ALI or ARDS. Do the authors believe this approach will reverse of stabilize disease, when treatment is delivered after the condition is shown to exist? Some discussion of the clinical use of the drug product should be included.
Response: We thank the reviewer for this insightful and clinically relevant question. The reviewer rightly points out that our current study employed a prophylactic administration regimen (NACOS was given prior to LPS challenge) to evaluate its protective potential against ALI. We agree that this model is primarily suited for establishing proof-of-concept and mechanistic efficacy rather than directly mimicking the clinical scenario, where treatment is initiated after the onset of symptoms.
Although administered prophylactically, the potent anti-inflammatory effects observed in our study—such as the significant reduction in inflammatory cell infiltration, the suppression of pro-inflammatory cytokines, and the restoration of alveolar architecture (Fig. 1A, 1D and Fig. 2A-2H)—suggest that NACOS doesn’t merely prevent injury but may also actively promote the resolution of inflammation. This key property is crucial for therapeutic agent aimed at treating established ALI. Furthermore, the upregulation of tight junction proteins (Claudin-1) and epithelial sodium channel (γ-EnaC) following NACOS treatment (Fig. 1B, 1C, 1E and 1F) indicates its role in restoring epithelial barrier integrity and enhancing alveolar fluid clearance. These processes are fundamental to reversing pulmonary edema in patients with existing ALI/ARDS. Therefore, it is plausible that NACOS could exert therapeutic benefits even when administered after disease onset by facilitating these repair mechanisms.
Moreover, modern research has revealed that NACOS possesses a range of potent biological activities, including antitumor (doi: 10.1016/j.carbpol.2014.04.102), immunomodulatory (doi: 10.3390/md18080421), and anti-inflammatory effects (doi: 10.3892/etm.2024.12600), and it has been recognized as an excellent candidate for a human health dietary supplement (doi: 10.3390/md18080421). This broad pharmacological profile further supports its potential for therapeutic application beyond mere prophylaxis.
We acknowledge that the definitive answer to the reviewer’s excellent question requires a dedicated therapeutic interventional study where NACOS is administered after LPS challenge. This is a recognized limitation of our current work and a primary objective for our subsequent research. To address these points, we have expanded the Discussion section (Lines 454–470) in the revised manuscript.
Comment 3. Lin 196 and 197-What are the upregulated metabolites indicative of mechanistically? Does their upregulation also correlate with recovery of the lungs?
Response: We sincerely appreciate this valuable comment. The upregulation of these specific metabolites is indicative of several potential mechanistic pathways through which NACOS may exert its protective effects:
(1) γ-Glu-Leu. This dipeptide is an intermediate in the glutathione (GSH) metabolic pathway. GSH is the most crucial intracellular antioxidant. Its marked upregulation strongly suggests an activation of the glutathione metabolism. This indicates that NACOS administration may potentiate cellular antioxidant defenses, thereby mitigating oxidative stress by neutralizing reactive oxygen species – a core pathological component in many lung injuries (DOI: 10.1186/s11658-022-00318-8,10.1007/s00253-021-11429-1).
(2) Agomelatine. This compound is a known melatonin receptor agonist and 5-HT2C receptor antagonist. Importantly, melatonin and its analogues exhibit potent anti-inflammatory and antioxidant properties. The upregulation of agomelatine suggests that NACOS treatment may activate melatonin-associated signaling pathways, consequently modulating inflammatory responses and reducing oxidative damage (DOI: 10.1007/s00210-023-02754-5).
(3) p-acetamidophenyl-β-D-glucuronide. This is a Phase II metabolite of the drug acetaminophen, generated through catalysis by UDP-glucuronosyltransferase enzymes. Its increased levels indicate a potential enhancement of xenobiotic detoxification pathways, specifically glucuronidation and acetylation (DOI: 10.1186/s12967-023-04499-4, 10.1152/ajplung.00072.2021). This implies that NACOS may facilitate the clearance of harmful substances or metabolic waste products, reducing their toxic burden on lung tissue.
Overall, the upregulation of these metabolites suggests that NACOS treatment enhances antioxidant defenses, modulates anti-inflammatory pathways, and promotes detoxification processes, collectively contributing to the mitigation of lung injury. Therefore, we believe that the upregulation of these metabolites is not merely an accompanying change but is mechanistically linked to the positive effects of NACOS, contributing to the recovery of lung function through mitigating oxidative stress and inflammation.
Comment 4. Line 330 – Since alveolar macrophages play a significant role in phospholipid metabolism were they looked at as a unique cell type for the impact of NACOS? In all of the metabolomic studies were specific markers associated with macrophage activity?
Response: We thank the reviewer for this insightful and critical question regarding the role of alveolar macrophages (AMs), which are indeed the sentinel immune cells and master regulators of phospholipid homeostasis in the alveoli. In the present study, we did not isolate AMs for separate omics analysis, which is a limitation of our work. We agree that investigating the direct impact of NACOS on AMs at a single-cell-type resolution would be a fascinating and necessary future direction to precisely delineate the mechanism.
However, the profound suppression of pro-inflammatory factors (e.g., IL-1β, TNF-α, MCP-1) in lung tissue homogenates, as we reported, strongly suggests that NACOS modulates the functional state of macrophages residing in the lung (Fig. 2). More importantly, our untargeted metabolomics data provide compelling circumstantial evidence that AMs are likely a key cellular target of NACOS. This is based on the finding that glycerophospholipid metabolism emerged as the most significantly altered pathway (Fig. 4A and Fig. 7A ). Given that AMs are primarily responsible for the synthesis, secretion, and recycling of pulmonary surfactant (which is predominantly composed of glycerophospholipids), the significant perturbation of this pathway strongly implicates AM function in the therapeutic effects of NACOS.
Regarding specific metabolomic markers associated with macrophage activity, we appreciate the reviewer's guidance to delve deeper into our data. While we did not measure canonical macrophage polarization markers (e.g., CD86, Arg1) at the protein level, our metabolomics results revealed several alterations in pathways and metabolites that are hallmarks of macrophage immunometabolism:
(1) Glycerophospholipid / Ether lipid metabolism: As the most impacted pathway, its alteration is directly indicative of changes in membrane lipid composition and surfactant metabolism, a process quintessentially governed by AMs (DOI: 10.1186/s13567-025-01580-2, 10.1016/j.jbc.2022.101709). The shift in this pathway suggests NACOS may promote membrane repair and restore surfactant homeostasis, a key function of AMs.
(2) Arachidonic acid metabolism: This pathway is a classic source of pro-inflammatory (e.g., prostaglandins, leukotrienes) and pro-resolving lipid mediators (doi: 10.3390/nu14071414,10.1128/microbiolspec.MCHD-0035-2016). Its perturbation suggests NACOS may reprogram the lipid mediator profile secreted by macrophages, shifting the balance from inflammation towards resolution.
(3) Glycine, serine, and threonine metabolism / Cysteine and methionine metabolism: These amino acid metabolism pathways are crucial for generating methyl donors and glutathione synthesis, directly linking to the antioxidant capacity and functional polarization of macrophages (doi: 10.3389/fimmu.2021.762564, 10.3389/fimmu.2022.780839,10.1016/j.redox.2024.103123).
Upon re-examining our metabolite list, we identified several molecules whose levels are known to be influenced by or to influence macrophage activity,such as LysoPC(22:2)、LysoPC(20:5)、Leukotriene C4、MG (18:1/0:0/0:0), PG (18:2/18:1), PG (18:0/18:2), PE (P-16-0/22:4), LysoPE (0:0/14:0), and LysoPE (14:1/0:0), among others (Supplementary table).
In conclusion, while we lack direct cellular evidence, the strong modulation of glycerophospholipid metabolism—a pathway intrinsically linked to AM function—combined with the changes in other immunometabolic pathways, provides a strong rationale that NACOS exerts its effects, at least in part, by reprogramming macrophage lipid metabolism and function. We are incredibly grateful for this suggestion and will explicitly focus on AMs in our subsequent mechanistic studies. Please refer to lines 427-447 for more details.

Reviewer 2 Report
Comments and Suggestions for Authors
In this manuscript entitled “Glycerophospholipid metabolism modulation by NACOS alleviates pulmonary inflammation in murine acute lung injury”, the authors investigated the protective effects of N-Acetylchitooligosaccharides (NACOS) in a murine model of ALI induced by LPS. They also explored the alteration of metabolism as a potential mechanism. They found that NACOS pretreatment attenuates LPS induced lung inflammation and significantly altered glycerophospholipid metabolism. They also conclude that the identified metabolic alterations may serve as potential biomarkers for the progression of ALI and for monitoring therapeutic interventions. Overall, the study is designed logically, and the conclusions are well supported by the data. Identification of altered glycerophospholipid metabolism in ALI model is novel. However, the major concerns are from the scientific rigors and reproducibility of the experiments.
- Sex as a biological variable was not considered, as only male mice were included in this study.
- NACOS was administered in drinking water for two weeks prior to LPS treatment. Therefore, it is more appropriate to describe its effect as protective and preventive rather than therapeutic.
- Endpoint evaluation was performed 8 hours after LPS treatment, which represents a very early stage of ALI. To fully understand the effects of NACOS, it would be important to include longer time points, extending up to 14 days.
- Please use dot plot to indicate the sample size in each experiment.
- The figure legends are overly concise and lack critical information, such as treatment dose and duration.
- Please quantify Figure 1A, 1B and 1C.
Author Response
Dear Editor,
On behalf of all the contributing authors, I highly appreciate your letter and reviewers’ constructive comments concerning our article entitled “Glycerophospholipid metabolism modulation by NACOS alleviates pulmonary inflammation in murine acute lung injury” (ID: ijms-3808091). These comments are very pertinent and valuable for improving our paper. According to the reviewer’s suggestions, we have revised the manuscript. The revision details were marked in red. If you have other suggestions for mending the paper, we would greatly appreciate your comments. The responses to reviewers’ comments are listed below.
Comments from the editors and reviewers:
-Reviewer 2
Reviewer #2: In this manuscript entitled “Glycerophospholipid metabolism modulation by NACOS alleviates pulmonary inflammation in murine acute lung injury”, the authors investigated the protective effects of N-Acetylchitooligosaccharides (NACOS) in a murine model of ALI induced by LPS. They also explored the alteration of metabolism as a potential mechanism. They found that NACOS pretreatment attenuates LPS induced lung inflammation and significantly altered glycerophospholipid metabolism. They also conclude that the identified metabolic alterations may serve as potential biomarkers for the progression of ALI and for monitoring therapeutic interventions. Overall, the study is designed logically, and the conclusions are well supported by the data. Identification of altered glycerophospholipid metabolism in ALI model is novel. However, the major concerns are from the scientific rigors and reproducibility of the experiments.
Comment 1. Sex as a biological variable was not considered, as only male mice were included in this study.
Response: We sincerely thank the reviewer for highlighting this important point. We completely agree that considering sex as a biological variable is crucial for the rigor and generalizability of biomedical research. Our decision to use male mice in this initial study was primarily to first establish a clear and stable model of ALI while controlling for potential confounding variables, including the hormonal fluctuations associated with the female estrous cycle (doi: 10.1177/1747493018778713).
We acknowledge that this choice is a limitation of the present study. Importantly, we are fully committed to investigating sex-dependent differences in our follow-up work. We believe this will significantly strengthen the impact of our research program. Thank you again for this constructive suggestion.
Comment 2. NACOS was administered in drinking water for two weeks prior to LPS treatment. Therefore, it is more appropriate to describe its effect as protective and preventive rather than therapeutic.
Response: Thank you for your comment. We completely agree with the reviewer that based on our experimental design, where NACOS was administered prior to the LPS challenge, the effects observed are indeed preventive and protective rather than therapeutic. The use of the term “therapeutic” was inaccurate in this context. We have now carefully revised the manuscript to replace any inappropriate instances of “therapeutic” or related terms with the more accurate descriptors “protective” and “prophylactic” throughout the text, including in the abstract, results, and discussion sections. The revision details were marked in red.
Comment 3. Endpoint evaluation was performed 8 hours after LPS treatment, which represents a very early stage of ALI. To fully understand the effects of NACOS, it would be important to include longer time points, extending up to 14 days.
Response: Thank you for your suggestion. We completely agree that the inclusion of longer time points is crucial for comprehensively understanding the dynamic effects of NACOS on the entire spectrum of ALI, from the acute inflammatory phase through the resolution and repair (or potential fibrotic) phases. Our current study was primarily designed to investigate the early protective mechanisms of NACOS, focusing on the initial inflammatory burst that occur within the first hours after LPS challenge. The 8-hour time point was selected to optimally capture these key early events, as suggested by previous literature (doi: 10.1155/2022/2686992, 10.1080/13880209.2016.1216132).
However, we fully acknowledge the reviewer’s point that this early time point represents a limitation of our present work, as it does not allow us to assess the long-term outcomes of NACOS treatment, such as its impact on resolution of inflammation, tissue repair processes, and potential fibrotic progression.
In response to this excellent suggestion, we have now added a statement to the Discussion section (Lines 448-454) of the revised manuscript to explicitly acknowledge this limitation and highlight the importance of future studies with extended time courses.
Comment 4. Please use dot plot to indicate the sample size in each experiment.
Response: We thank the reviewer for this constructive suggestion. We agree that overlaying individual data points is essential for indicating the sample size and showing the data distribution clearly.
In response, we have now revised all relevant bar graphs in the manuscript to include superimposed dot plots that show each individual data point. This modification has been applied to the following figures: figure 1 (e.g., figure 1D, 1E, 1F, 1G), figure 2 (e.g., Figure 2A to 2H), figure 5 (e.g., figure 5B to 5I) and figure 8 (e.g., figure 8B to 8J).
Comment 5. The figure legends are overly concise and lack critical information, such as treatment dose and duration.
Response: We sincerely thank the reviewer for this critical and constructive comment. We fully agree that providing detailed experimental information in the figure legends is essential for the clarity and reproducibility of our work.
In response to this suggestion, we have thoroughly revised all relevant figure legends throughout the manuscript to include comprehensive details regarding treatment doses, durations, and other key methodological parameters. The revision details were marked in red (Lines 116-121, 140-145, 183-188, 219-224, 241-246, 271-276, 302-306 and 323-328).
Comment 6. Please quantify Figure 1A, 1B and 1C.
Response: We thank the reviewer for this critical suggestion to improve the objectivity and rigor of our data. We completely agree that quantitative analysis is essential. In response, we have now performed comprehensive quantitative analyses for the data presented in Figures 1A, 1B, and 1C. These new quantitative results have been added to the manuscript (Figures 1D, 1E, and 1F).
The statistical analysis confirms the significant differences between groups that we initially observed qualitatively, thereby greatly strengthening our conclusions. Thank you once again for this constructive suggestion.
Round 2
Reviewer 2 Report
Comments and Suggestions for Authors
The overall quality has been improved. However, several concerns have not been addressed.
- Sex as a biological variable was not addressed in this study. This should be discussed as a study limitation at least.
- The therapeutic effect of NACOS was not evaluated in the study. Therefore, it is misleading to use the terms “therapeutic” or “therapy.” Please carefully revise the wording in lines 36, 106, and 520 accordingly.
Author Response
Dear Editor,
On behalf of all the contributing authors, I highly appreciate your letter and reviewers’ constructive comments concerning our article entitled “Glycerophospholipid metabolism modulation by NACOS alleviates pulmonary inflammation in murine acute lung injury” (ID: ijms-3808091). These comments are very pertinent and valuable for improving our paper. According to the reviewer’s suggestions, we have revised the manuscript. The revision details were marked in yellow. If you have other suggestions for mending the paper, we would greatly appreciate your comments. The responses to reviewers’ comments are listed below.
Comments from the editors and reviewers:
-Reviewer 3
Reviewer #3: The overall quality has been improved. However, several concerns have not been addressed.
Comment 1. Sex as a biological variable was not addressed in this study. This should be discussed as a study limitation at least.
Response: We sincerely thank the reviewer for highlighting this important point. We have now addressed the limitation regarding the sex of the mice used in this study by adding a corresponding statement in the Discussion section of the revised manuscript. Please refer to lines 454-458 for more details.
Comment 2. The therapeutic effect of NACOS was not evaluated in the study. Therefore, it is misleading to use the terms “therapeutic” or “therapy.” Please carefully revise the wording in lines 36, 106, and 520 accordingly.
Response: We sincerely thank the reviewer for this critical and accurate comment. We fully agree that the term 'therapeutic' is not appropriate for our study design, in which NACOS was administered prophylactically prior to LPS challenge. We apologize for this oversight and any misunderstanding it may have caused. We have carefully revised the manuscript to replace 'therapeutic' or 'therapy' with more precise terms such as 'protective,' 'protective effect,' 'prophylactic effect,' or 'preventive effect' throughout the manuscript, as specifically suggested in lines 37, 110, and 541. The changes have been made to accurately reflect the preventive nature of our study.